# Respiratory virus behavior: Results of laboratory-based epidemiological surveillance

**Porfirio Felipe Hernández Bautista**[1⊙], **David Alejandro Cabrera Gaytán**[1⊙]*,
**Alfonso Vallejos Parás**[2⊙], **Alejandro Moctezuma Paz**[3⊙], **Clara Esperanza Santacruz Tinoco**[1‡], **Julio Elias Alvarado Yaah**[1⊙], **Yu Mei Anguiano Hernández**[1‡], **Bernardo Martínez Miguel**[1⊙], **Lumumba Arriaga Nieto**[2⊙], **Leticia Jaimes Betancourt**[4‡], **Nancy Sandoval Gutiérrez**[1‡]

**1** Instituto Mexicano del Seguro Social, Coordinación de Calidad de Insumos y Laboratorios Especializados, México, México, **2** Instituto Mexicano del Seguro Social, Coordinación de Vigilancia Epidemiológica, México, México, **3** Instituto Mexicano del Seguro Social, Coordinación de Investigación en Salud, México, México, **4** Instituto Mexicano del Seguro Social, Unidad de Medicina Familiar No. 7, México, México

⊙ These authors contributed equally to this work.
‡ CEST, YMAH, LJB and NSG also contributed equally to this work.
* david.cabrerag@imss.gob.mx, dcpreventiva@gmail.com

**Data Availability Statement:** Data can be found at the following repository: 10.6084/m9.figshare. 24895014.

## Abstract

### Background

Respiratory viruses have clinical and epidemiological importance. With the COVID-19 pandemic, interest has focused on SARS-CoV-2, but as a result, the number of samples available for the differential diagnosis of other respiratory viruses has increased.

### Study design

Cross-sectional study.

### Objective

To describe the epidemiological behavior of respiratory viruses based on a laboratory-based epidemiological surveillance system using data from 2017 to 2023.

### Methods

Univariate, bivariate and multivariate analyses of data from a laboratory database of respiratory viruses detected by multiplex RT–qPCR were performed.

### Results

A total of 4,632 samples with positive results for at least 1 respiratory virus, not including influenza or SARS-CoV-2, were analyzed. The most common virus detected was respiratory syncytial virus in 1,467 (26.3%) samples, followed by rhinovirus in 1,384 (24.8%) samples. Most of the samples were from children under 5 years of age. The age-adjusted odds ratio (OR) of death for patients infected with parainfluenza virus 4 was 4.1 (95% confidence interval [95% CI] 2.0–8.2).

**Funding:** The author(s) received no specific funding for this work.

**Competing interests:** The authors have declared that no competing interests exist.

## Conclusion

Respiratory syncytial virus and rhinovirus had the highest frequency and proportion of coinfections, whereas parainfluenza virus 4 was associated with an increased risk of death.

## Background

The main known respiratory viruses cause the common cold, conjunctivitis and enteritis and have been associated with outbreaks depending on the identified type of virus. Usually, these conditions are self-limiting and do not cause major complications, except in people with immunodeficiency [1]. Furthermore, they can occur as coinfections with other respiratory viruses and manifest clinically as influenza-like illness [2].

Since 2020, laboratory detection of SARS-CoV-2 and, therefore, genomic surveillance have been increasingly relevant. Cases of acute childhood hepatitis were reported in 2022 in the United Kingdom, where agents such as adenovirus, SARS-CoV-2 and coinfection with these viruses were identified by laboratory testing [3–5]. Similarly, the United States [6], Canada [7], and Europe [8], especially the United Kingdom [4], have shown an increase in the number of samples collected and in positive adenovirus results.

Surveillance of respiratory viruses in Mexico has been carried out using data from the Influenza Epidemiological Surveillance System from 2009–2020 [9] and the Viral Respiratory Disease Epidemiological Surveillance System (ERV) from 2020 to the present with new COVID-19 variables [10] as a nonprobabilistic sampling of negative results for influenza or SARS-CoV-2. In both periods, samples were analyzed with different operational case definitions but in a syndromic manner to identify mild and severe cases of acute respiratory disease. The Mexican Institute for Social Security (*Instituto Mexicano del Seguro Social*, IMSS) has the diagnostic capacity to identify 14 respiratory viruses in addition to influenza and SARS-CoV-2 in the network's reference laboratories with the endorsement of the diagnostic competence of the Ministry of Health [11].

It is of interest to describe the epidemiological behavior of respiratory viruses other than the influenza virus and SARS-CoV-2, given that there is little diffusion of those viruses in the population served by social security, and to report the results of the laboratory-based surveillance at the IMSS.

## Materials and methods

### Study overview

A descriptive study based on epidemiological laboratory surveillance was carried out. The study was retrolective, multicenter and nationally based. The data were sourced from the laboratory-based epidemiological surveillance system, which operates in all IMSS medical units; if a patient meets the operational definition of a case of influenza-like illness, severe acute respiratory infection or suspected case of viral respiratory disease, it is required to collect and process a sample of pharyngeal exudate, nasal exudate or both according to the standardized guidelines in Mexico [9, 11]. In sentinel units for the surveillance of viral respiratory disease, of the total cases registered in the epidemiological surveillance system, at least 10% of cases negative for SARS-CoV-2 and for influenza are processed for other respiratory viruses; likewise, in nonsentinel units, samples are processed for other respiratory viruses in 10% of severe cases and 100% of deaths. The results of laboratory samples processed for other respiratory viruses that were registered in the information system for epidemiological surveillance by the

Institute's laboratory between January 2017 and January 2023 were utilized in this study. The diagnosis of respiratory viruses was achieved by multiplex RT–qPCR (FilmArray™) using a panel of 14 respiratory viruses. The variables that were collected for each sample were state, age, sex, type of care, date of result and result, according to the standardized guidelines in Mexico [9, 11].

A trend analysis was carried out over time, by age group stratified by state and by respiratory virus coinfections by age group, not including results for influenza viruses and SARS-CoV-2. The positivity rate was estimated by year, type of virus and state.

## Laboratory analysis of samples

The SuperScript™ III Platinum® One-Step RT–qPCR System Kit (Invitrogen, Carlsbad, California, USA. Catalog: 12574035) was used with a 7500 Fast Real-Time PCR System (Applied Biosystems®, Foster City, California, USA) to amplify viral genetic material. Primers and probes were used for each of the following viruses: metapneumovirus, respiratory syncytial virus, human parainfluenza viruses 1–4, human coronavirus (OC43, HKU1, 229E, NL63), rhinovirus, enterovirus, adenovirus and bocavirus. The human RNAse P (RP) gene was used as an internal control. The samples were evaluated in uniplex reactions with the following reaction mixture: 12.5 μL of 2x reaction mix, 0.5 μL of each primer and probe, 1 μL of enzyme, 5.5 μL of RNase-free water, and 5 μL of total nucleic acid. The following thermocycling conditions were used: one cycle of 45˚C for 10 min and 95˚C for 10 min and 45 cycles of 95˚C for 15 sec and 55˚C for 1 min. DNA or RNA lyophilizates (AmpliRun® Vircell; Granada–Spain) were used as the positive controls for all evaluated viruses. Samples that presented amplification for any of the viral markers plus the internal control were considered positive. Samples without amplification of the viral markers but with amplification of the internal control were considered negative.

The primers and probes used for the afore mentioned diagnoses and the amplification program were described at S1 File.

The reagents were used in accordance with the manufacturers' instructions and in accordance with good molecular biology laboratory practices and the Quality Management System in ISO 9001:2025 of the Central Epidemiology Laboratory.

## Statistical analysis

For the univariate analysis, simple frequencies and measures of central tendency and dispersion were determined, and a time series analysis was carried out to determine correlations over time with respect to age, in addition to a chi square trend. Multivariate analysis of the associations between odds ratios (ORs) and variables and nonconditional logistic regression were also performed.

## Ethical considerations

Since this was a retrospective and retrolective study with no direct intervention with the selected subject or with a medical unit operation, informed consent was not required. All participant data were handled confidentially through the folio number. This study was approved by Committee No. 3605 (approval registration R-2022-3605-057).

## Resources, financing and feasibility

The acquisition, concentration, validation and analysis of the "Epidemiological Control System for Laboratory" ["Sistema de Control Epidemiológico para Laboratorio (SISCEP)]

database were performed with computer equipment from the IMSS, according to the user profiles. The financing was its own.

## Results

A total of 4,632 samples from patients treated at the IMSS medical units between January 2017 and January 2023 with a positive result for at least 1 respiratory virus were analyzed. Of the 1,691,678 samples registered with the diagnostic algorithm for respiratory disease, 26,483 were processed for the identification of other respiratory viruses (1.6%); since 2020 (proportion less than 10% of what is regulated in the country), the number of samples has increased. A total of 5,576 positive results were identified from 4,632 samples.

During 2022, there was a greater number of samples, and therefore, the identification of viral agents increased. Compared to 2021, the number of adenoviruses increased more than 4-fold, with a total of 198 positive results.

Fourteen types of respiratory viruses were evaluated, including adenovirus; respiratory syncytial virus; rhinovirus; metapneumovirus; parainfluenza types 1, 2, 3, and 4; bocavirus; enterovirus; coronavirus OC43; coronavirus 229E; coronavirus NL63; and coronavirus HKU1, for a total of 5,576 positive results. The most frequently detected virus was respiratory syncytial virus in 1,467 (26.3%) samples, followed by rhinovirus in 1,384 (24.8%) samples (Table 1). It can be seen that during 2020 and after this year, surveillance for other respiratory viruses increased (Fig 1). Adenoviruses ranked sixth in frequency, with 383 positive results (6.8%). In 16.9% of the samples (*n* = 785), more than one viral agent was identified, and 5 different viruses were present in 4 samples. Infection with a single virus was more common in males under 5 years of age and over 65 years of age, whereas coinfection with 2 viruses was more common in females aged 15 years or older (Table 2). The most frequent coinfection was respiratory syncytial virus with rhinovirus in 82 patients. Adenovirus occurred simultaneously with all identified respiratory viruses, except coronavirus 229E. The viruses that were identified in combination with adenoviruses most often were respiratory syncytial virus and rhinovirus. In terms of age, the different viral types were mostly identified in patients under 5 years of age and patients between 25 and 44 years of age. Additionally, in children under 5 years of age, the detection of more than one virus was more common (Table 2). Between 2017 and 2019, the

**Table 1. Respiratory viruses identified from 2017 to 2023.**

| VIRUS | TOTAL | % | 2017 | 2018 | 2019 | 2020 | 2021 | 2022 | 2023 |
|---|---|---|---|---|---|---|---|---|---|
| Respiratory syncytial virus | 1,467 | 26.31 | 38 | 19 | 120 | 35 | 312 | 865 | 78 |
| Rhinovirus | 1,384 | 24.82 | 14 | 28 | 62 | 197 | 406 | 592 | 85 |
| Metapneumovirus | 566 | 10.15 | 4 | 47 | 21 | 28 | 63 | 359 | 44 |
| Parainfluenza virus 3 | 484 | 8.68 | 2 | 11 | 18 | 35 | 157 | 226 | 35 |
| Bocavirus | 390 | 6.99 | 3 | 7 | 17 | 31 | 73 | 238 | 21 |
| Adenovirus | 383 | 6.87 | 7 | 14 | 31 | 60 | 46 | 198 | 27 |
| Enterovirus | 250 | 4.48 | 6 | 2 | 9 | 15 | 15 | 194 | 9 |
| Coronavirus OC43 | 166 | 2.98 | 3 | 5 | 11 | 17 | 45 | 65 | 20 |
| Coronavirus 229E | 125 | 2.24 | 3 | 3 | 3 | 12 | 57 | 40 | 7 |
| Parainfluenza virus 4 | 116 | 2.08 | 1 | 6 | 5 | 19 | 19 | 65 | 1 |
| Coronavirus NL63 | 111 | 1.99 | 2 | 2 | 5 | 60 | 15 | 22 | 5 |
| Parainfluenza virus 1 | 54 | 0.97 | 2 | 5 | 8 | 11 | 3 | 21 | 4 |
| Coronavirus HKU1 | 52 | 0.93 | 0 | 4 | 4 | 12 | 8 | 24 | 0 |
| Parainfluenza virus 2 | 28 | 0.50 | 3 | 2 | 6 | 6 | 2 | 8 | 1 |
| **TOTAL** | **5,576** | **100.00** | **88** | **155** | **320** | **538** | **1,221** | **2,917** | **337** |

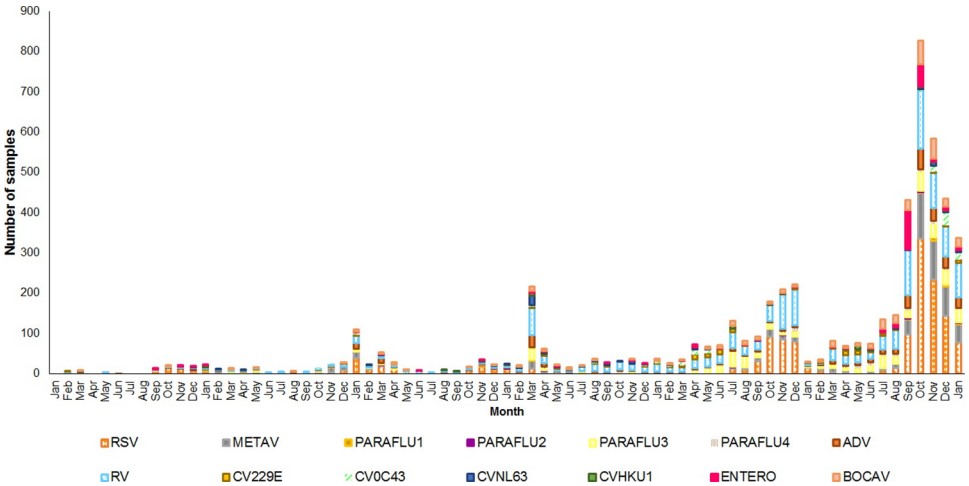

**Fig 1. Behavior of respiratory viruses by month, 2017–2023.**

positivity rate was low for most viruses; however, during the COVID-19 pandemic, these rates were more frequent and greater. The occurrence of rhinovirus and respiratory syncytial virus was more common, especially from 2020 to 2023. Moreover, Mexico City was the one region with positive samples throughout the study period. The highest positive rate for rhinovirus occurred in Aguascalientes (100% in 2011), while the highest positive rate for respiratory syncytial virus occurred in Michoacán (62.2% in 2021), Oaxaca (54.7% in 2022) and Tamaulipas (47.3% in 2022). Metapneumovirus was more prevalent in Tamaulipas (40.0% in 2020 and 47.0% in 2023), and Parainfluenza 3 was more prevalent in Sinaloa (66.7% in 2018), (Fig 2).

When the samples were stratified by sex, more samples were determined to come from males than from females, although the difference was not statistically significant (Table 3). In terms of the type of medical care, the majority of patients received hospital care. Lethality was greater in patients with a single infection of parainfluenza 4 or parainfluenza 1 virus (8.6% and 7.4%, respectively) (Table 4). No deaths were reported in patients who were positive for enterovirus alone.

ORs were calculated to determine the association between age and the presence of different viral agents. For adenoviruses, the OR showed that the older the patient was, the lower the risk was, with a statistically significant trend (chi square$_{trend}$ = 37.0; $p < 0.001$). The same was true

**Table 2. Number of simultaneous virus infections by age.**

| AGE GROUP | 1 virus | | 2 virus | | 3 virus | | 4 virus | | 5 virus | |
|---|---|---|---|---|---|---|---|---|---|---|
| | W | M | W | M | W | M | W | M | W | M |
| Less than 1 year | 296 | 441 | 69 | 86 | 12 | 24 | 0 | 0 | 0 | 1 |
| 1–4 years | 443 | 555 | 165 | 155 | 33 | 39 | 3 | 8 | 1 | 1 |
| 5–9 years | 164 | 164 | 16 | 30 | 5 | 3 | 1 | 0 | 0 | 0 |
| 10–14 years | 63 | 66 | 6 | 7 | 0 | 0 | 0 | 0 | 0 | 0 |
| 15–24 years | 101 | 68 | 11 | 6 | 0 | 0 | 0 | 0 | 0 | 0 |
| 25–44 years | 336 | 211 | 26 | 18 | 3 | 1 | 0 | 0 | 1 | 0 |
| 45–64 years | 239 | 199 | 15 | 13 | 0 | 0 | 0 | 0 | 0 | 0 |
| 65 and + years | 243 | 258 | 14 | 12 | 0 | 0 | 0 | 0 | 0 | 0 |
| **Total** | **1,885** | **1,962** | **322** | **327** | **53** | **67** | **4** | **8** | **2** | **2** |

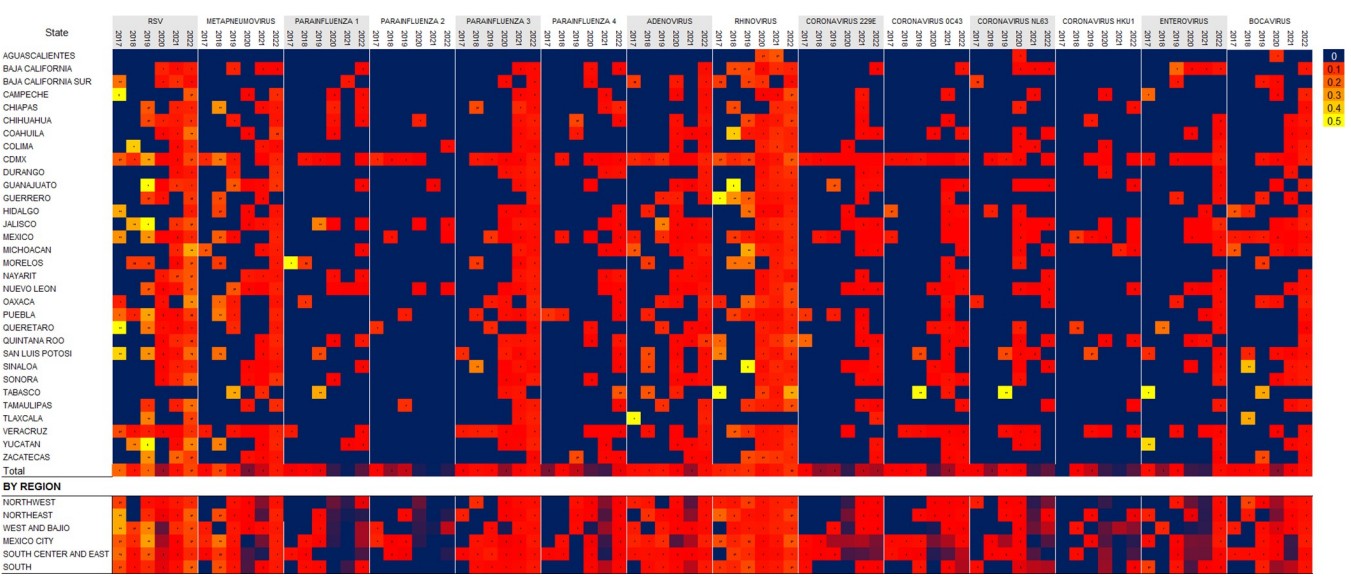

**Fig 2. Heat graph of positive results for respiratory viruses from 2017 to 2022.**

for respiratory syncytial virus (chi square$_{trend}$ = 717.44; $p < 0.001$) and bocavirus (chi square$_{trend}$ = 81.3; $p < 0.001$). However, infection with coronavirus 229E had an increased probability in older patients (chi square$_{trend}$ = 89.0; $p < 0.001$) (Table 5). When sex was added to the logistic regression analysis, the same trends remained, while female or male sex did not show any association with the identification of any type of virus (Table 6).

The risk of death associated with the virus was not significant for adenovirus; however, the risk of death was lower for respiratory syncytial virus (hazard ratio [HR] = 0.53, 95%

**Table 3. Presentation of respiratory virus infection by age group and sex.**

| AGE GROUP | RESPIRATORY SYNCYTIAL VIRUS | | RHINOVIRUS | | METAPNEUMOVIRUS | | PARAINFLUENZA 3 | | BOCAVIRUS | | ADENOVIRUS | | ENTEROVIRUS | |
|---|---|---|---|---|---|---|---|---|---|---|---|---|---|---|
| | W | M | W | M | W | M | W | M | W | M | W | M | W | M |
| Less than 1 year | 229 | 327 | 67 | 97 | 49 | 55 | 40 | 61 | 32 | 55 | 26 | 37 | 12 | 18 |
| 1–4 years | 280 | 286 | 163 | 196 | 98 | 101 | 62 | 79 | 89 | 95 | 85 | 111 | 50 | 71 |
| 5–9 years | 39 | 45 | 60 | 75 | 37 | 38 | 18 | 12 | 15 | 8 | 9 | 22 | 23 | 16 |
| 10–14 years | 9 | 11 | 29 | 33 | 6 | 7 | 8 | 11 | 2 | 3 | 1 | 6 | 7 | 2 |
| 15–24 years | 12 | 11 | 50 | 33 | 5 | 3 | 8 | 3 | 8 | 7 | 5 | 5 | 7 | 3 |
| 25–44 years | 46 | 24 | 147 | 90 | 30 | 24 | 49 | 17 | 13 | 7 | 19 | 14 | 7 | 9 |
| 45–64 years | 36 | 30 | 85 | 79 | 26 | 24 | 32 | 28 | 20 | 8 | 9 | 7 | 7 | 2 |
| 65 and + years | 38 | 44 | 90 | 90 | 40 | 23 | 26 | 30 | 9 | 19 | 12 | 15 | 8 | 8 |
| **Total** | **689** | **778** | **691** | **693** | **291** | **275** | **243** | **241** | **188** | **202** | **166** | **217** | **121** | **129** |
| AGE GROUP | CORONAVIRUS OC43 | | CORONAVIRUS 229E | | PARAINFLUENZA 4 | | CORONA NL63 | | PARAINFLUENZA 1 | | CORONAVIRUS HKU1 | | PARAINFLUENZA 2 | |
| | W | M | W | M | W | M | F | M | W | M | W | M | W | M |
| Less than 1 year | 5 | 18 | 2 | 2 | 5 | 10 | 3 | 3 | 0 | 3 | 1 | 0 | 3 | 4 |
| 1–4 years | 23 | 18 | 1 | 9 | 10 | 20 | 11 | 9 | 6 | 16 | 8 | 8 | 1 | 5 |
| 5–9 years | 2 | 3 | 3 | 4 | 5 | 4 | 3 | 4 | 2 | 0 | 1 | 1 | 1 | 2 |
| 10–14 years | 3 | 1 | 4 | 2 | 0 | 0 | 0 | 1 | 0 | 2 | 0 | 0 | 0 | 1 |
| 15–24 years | 2 | 2 | 6 | 3 | 6 | 1 | 8 | 1 | 1 | 2 | 1 | 3 | 1 | 2 |
| 25–44 years | 27 | 17 | 19 | 12 | 8 | 9 | 20 | 13 | 4 | 3 | 9 | 7 | 2 | 2 |
| 45–64 years | 9 | 8 | 18 | 13 | 11 | 7 | 10 | 8 | 2 | 4 | 4 | 5 | 1 | 1 |
| 65 and + years | 15 | 13 | 14 | 13 | 7 | 7 | 7 | 10 | 4 | 5 | 0 | 4 | 1 | 1 |
| **Total** | **86** | **80** | **67** | **58** | **58** | **58** | **62** | **49** | **19** | **35** | **24** | **28** | **10** | **18** |

**Table 4. Type of care for respiratory virus.**

| VIRUS | AMBULATORY | HOSPITALIZED | DEATH | TOTAL | FATALITY RATE (%) |
|---|---|---|---|---|---|
| Parainfluenza virus 4 | 36 | 70 | 10 | 116 | 8.6 |
| Parainfluenza virus 1 | 14 | 36 | 4 | 54 | 7.4 |
| Parainfluenza virus 2 | 8 | 18 | 2 | 28 | 7.1 |
| Coronavirus 229E | 31 | 86 | 8 | 125 | 6.4 |
| Coronavirus HKU1 | 19 | 31 | 2 | 52 | 3.8 |
| Coronavirus OC43 | 56 | 106 | 4 | 166 | 2.4 |
| Adenovirus | 108 | 267 | 8 | 383 | 2.1 |
| Bocavirus | 91 | 291 | 8 | 390 | 2.1 |
| Rhinovirus | 417 | 939 | 28 | 1,384 | 2.0 |
| Coronavirus NL63 | 44 | 65 | 2 | 111 | 1.8 |
| Metapneumovirus | 93 | 463 | 10 | 566 | 1.8 |
| Respiratory syncytial virus | 290 | 1,157 | 20 | 1,467 | 1.4 |
| Parainfluenza virus 3 | 102 | 376 | 6 | 484 | 1.2 |
| Enterovirus | 90 | 160 | 0 | 250 | 0.0 |
| **TOTAL** | **1,399** | **4,065** | **112** | **5,576** | **2.0** |
| More than one virus | 245 | 531 | 9 | 785 | 1.1 |

confidence interval [95%CI] 0.32–0.86) and greater for parainfluenza virus 4 and 1 (HR = 4.01, 95%CI: 2.14–7.53; HR = 3.49, 95%CI: 1.33–9.16, respectively) (Table 7). In the multivariate logistic regression analysis, only the risk for parainfluenza remained.

The presence of 2 or more simultaneous viruses was determined to be a protective factor against death according to the bivariate analysis (HR = 0.47, 95% CI 0.24–0.94). However, there were a greater number of hospitalized patients with multiple viruses. This association disappeared in the multivariate analysis (Table 8).

## Discussion

A descriptive analysis of respiratory virus behavior was performed using samples from patients in medical units that serve the population with social security, which is captured by the epidemiological surveillance system. The most frequently detected viruses were respiratory syncytial virus and rhinovirus. Most of the samples processed since 2020 have focused on SARS-CoV-2 surveillance and, prior to that, on influenza; however, the present study focused on the differential diagnosis of other laboratory-identified respiratory viruses.

The frequency by type of respiratory virus, other than influenza or SARS-CoV-2, was similar to that reported in other parts of the world [12–16]. However, this finding contrasts with a study carried out in China during the COVID-19 pandemic, where between 2020 and 2021, the prevalence of adenovirus decreased considerably [17]. Additionally, in Iran, adenovirus had the lowest prevalence (0.1%) during the pandemic, whereas in our 2020 study, compared to the previous year, an increase was observed, which was continued in 2022 [18].

The increase in the number of samples for differential diagnosis of viruses during 2022 is believed to have been caused by the following:

1. The decrease in SARS-CoV-2 positivity in the interepidemic periods of the fourth and fifth waves, as well as during the sixth wave (3 epidemic waves occurred in 2022: the fourth wave from week 51 of 2021 to week 9 of 2022, which was very sudden; the fifth wave from week 22 to 33 of 2022; and the sixth from week 49 of 2022, which extended into 2023).

**Table 5. Raw odd ratio of the presence of respiratory virus with age.**

| AGE AND SEX | RESPIRATORY SYNCYTIAL VIRUS | | RHINOVIRUS | | METAPNEUMOVIRUS | | PARAINFLUENZA 3 | | BOCAVIRUS | | ADENOVIRUS | | ENTEROVIRUS | |
|---|---|---|---|---|---|---|---|---|---|---|---|---|---|---|
| | OR | 95%CI | OR | 95%CI | OR | 95%CI | OR | 95%CI | OR | 95%CI | OR | 95%CI | OR | 95%CI |
| Less than 1 year | 1 | -;- | 1 | -;- | 1 | -;- | 1 | -;- | 1 | -;- | 1 | -;- | 1 | -;- |
| 1–4 years | 0.67 | 0.62;0.73 | 1.44 | 1.22;1.70 | 1.26 | 1.01;1.58 | 0.92 | 0.72;1.17 | 1.4 | 1.10;1.78 | 2.06 | 1.57;2.76 | 2.67 | 1.80;3.94 |
| 5–9 years | 0.36 | 0.30;0.44 | 1.99 | 1.64;2.42 | 1.74 | 1.33;2.29 | 0.72 | 0.48;1.06 | 0.64 | 0.41;0.99 | 1.19 | 0.78;1.80 | 3.15 | 1.98;4.99 |
| 10–14 years | 0.23 | 0.15;0.35 | 2.47 | 1.95;3.12 | 0.81 | 0.47;1.41 | 1.23 | 0.77;1.94 | 0.37 | 0.15;0.91 | 0.72 | 0.33;1.55 | 1.96 | 0.95;4.04 |
| 15–24 years | 0.20 | 0.14;0.30 | 2.52 | 2.04;3.12 | 0.38 | 0.19;0.77 | 0.54 | 0.29;0.99 | 0.86 | 0.50;1.45 | 0.79 | 0.41;1.51 | 1.66 | 0.82;3.34 |
| 25–44 years | 0.19 | 0.15;0.24 | 2.25 | 1.89;2.67 | 0.80 | 0.59;1.10 | 1.01 | 0.76;1.36 | 0.35 | 0.22;0.57 | 0.81 | 0.54;1.22 | 0.83 | 0.45;1.51 |
| 45–64 years | 0.23 | 0.18;0.29 | 1.99 | 1.65;2.4 | 0.95 | 0.69;1.31 | 1.18 | 0.87;1.59 | 0.64 | 0.42;0.96 | 0.5 | 0.29;0.86 | 0.59 | 0.28;1.25 |
| 65 and + years | 0.26 | 0.21;0.31 | 1.93 | 1.61;2.32 | 1.06 | 0.79;1.43 | 0.97 | 0.71;1.33 | 0.56 | 0.37;0.85 | 0.75 | 0.48;1.17 | 0.94 | 0.51;1.70 |
| χ2 trend | test | p | test | p | test | p | test | p | test | p | test | p | test | p |
| | 539.00 | <0.001 | 91.00 | <0.001 | 5.67 | 0.02 | 0.77 | 0.37 | 42.00 | <0.001 | 37.00 | <0.001 | 21.00 | <0.001 |

| AGE AND SEX | CORONAVIRUS OC43 | | CORONAVIRUS 229E | | PARAINFLUENZA 4 | | CORONAVIRUS NL63 | | PARAINFLUENZA1 | | CORONAVIRUS HKU1 | | PARAINFLUENZA 2 | |
|---|---|---|---|---|---|---|---|---|---|---|---|---|---|---|
| | OR | 95%CI | OR | 95%CI | OR | 95%CI | OR | 95%CI | OR | 95%CI | OR | 95%CI | OR | 95%CI |
| Less than 1 year | 1 | -;- | 1 | -;- | 1 | -;- | 1 | -;- | 1 | -;- | 1 | -;- | 1 | -;- |
| 1–4 years | 1.18 | 0.71;1.95 | 1.65 | 0.52;5.26 | 1.32 | 0.71;2.44 | 2.20 | 0.88;5.47 | 4.85 | 1.45;16.17 | 10.59 | 1.40;79.75 | 0.56 | 0.19;1.68 |
| 5–9 years | 0.52 | 0.20;1.37 | 4.24 | 1.24;14.41 | 1.45 | 0.64;3.29 | 2.82 | 0.95;8.36 | 1.61 | 0.27;9.63 | 4.85 | 0.44;53.34 | 1.03 | 0.27;3.99 |
| 10–14 years | 1.13 | 0.39;3.24 | 9.81 | 2.80;34.34 | 2.63 | 1.03;6.63 | 1.09 | 0.13;8.99 | 4.36 | 0.73;25.87 | – | -;- | 0.93 | 0.11;7.54 |
| 15–24 years | 0.86 | 0.30;2.48 | 11.23 | 3.49;36.10 | 2.33 | 0.96;5.63 | 7.49 | 2.69;20.79 | 4.99 | 1.01;24.55 | 19.9 | 2.24;177.7 | 2.14 | 0.55;8.20 |
| 25–44 years | 2.98 | 1.82;4.88 | 12.08 | 4.28;34.04 | 1.76 | 0.88;3.51 | 8.57 | 3.61;20.33 | 3.63 | 0.94;14.01 | 24.93 | 3.31;187.57 | 0.89 | 0.26;3.02 |
| 45–64 years | 1.4 | 0.79;2.73 | 15.45 | 5.48;43.50 | 2.39 | 1.21;4.70 | 5.99 | 2.39;14.99 | 3.98 | 1.0;15.87 | 17.94 | 2.27;141.19 | 0.56 | 0.11;2.73 |
| 65 and + years | 2.14 | 1.24;3.68 | 11.89 | 3.18;33.82 | 0.47 | 0.15;1.43 | 4.99 | 1.98;12.59 | 5.28 | 1.43;19.44 | 7.05 | 0.79;62.92 | 0.5 | 0.1;2.24 |
| χ2 trend | test | p | test | p | test | p | test | p | test | p | test | p | test | p |
| | 18.00 | <0.001 | 89.00 | <0.001 | 5.78 | 0.02 | 36.00 | <0.001 | 2.40 | 0.12 | 9.05 | 0.00 | 0.11 | 0.07 |

**Table 6. Multivariate logistic regression analysis by age and sex associated with the presence of a respiratory virus.**

| Age and sex | RESPIRATORY SYNCYTIAL VIRUS | | RHINOVIRUS | | METAPNEUMOVIRUS | | PARAINFLUENZA 3 | | BOCAVIRUS | | ADENOVIRUS | | ENTEROVIRUS | |
|---|---|---|---|---|---|---|---|---|---|---|---|---|---|---|
| | OR | 95%CI | OR | 95%CI | OR | 95%CI | OR | 95%CI | OR | 95%CI | OR | 95%CI | OR | 95%CI |
| Man/Women | 0.91 | 0.8; 1.04 | 1.03 | 0.91; 1.18 | 0.85 | 0.71; 1.02 | 0.9 | 0.77; 1.1 | 1.0 | 0.77; 1.2 | 1.2 | 0.97; 1.5 | 1.0 | 0.75; 1.3 |
| Less than 1 year | 1.00 | –; – | 1.00 | –; – | 1.00 | –; – | 1.0 | –; – | 1.0 | –; – | 1.0 | –; – | 1.0 | –; – |
| 1–4 years | 0.45 | 0.4; 0.53 | 1.61 | 1.30; 1.97 | 1.30 | 1.01; 1.67 | 0.9 | 0.69; 1.2 | 1.5 | 1.11; 1.9 | 2.3 | 1.67; 3.0 | 2.8 | 1.87; 4.3 |
| 5–9 years | 0.18 | 0.14; 0.24 | 2.54 | 1.94; 3.33 | 1.90 | 1.37; 2.64 | 0.7 | 0.45; 1.1 | 0.6 | 0.38; 1.0 | 1.2 | 0.78; 1.9 | 3.4 | 2.07; 5.5 |
| 10–14 years | 0.11 | 0.06; 0.17 | 3.62 | 2.49; 5.26 | 0.78 | 0.43; 1.44 | 1.3 | 0.74; 2.1 | 0.4 | 0.14; 0.9 | 0.7 | 0.32; 1.6 | 2.0 | 0.93; 4.4 |
| 15–24 years | 0.09 | 0.05; 0.14 | 3.78 | 2.70; 5.29 | 0.34 | 0.16; 0.72 | 0.5 | 0.26; 1.0 | 0.8 | 0.47; 1.5 | 0.8 | 0.40; 1.6 | 1.7 | 0.81; 3.5 |
| 25–44 years | 0.08 | 0.06; 0.11 | 3.10 | 2.44; 3.93 | 0.76 | 0.54; 1.08 | 1.0 | 0.72; 1.4 | 0.3 | 0.20; 0.5 | 0.8 | 0.54; 1.3 | 0.8 | 0.44; 1.5 |
| 45–64 years | 0.11 | 0.08; 0.14 | 2.54 | 1.97; 3.28 | 0.93 | 0.65; 1.33 | 1.2 | 0.85; 1.7 | 0.6 | 0.39; 1.0 | 0.5 | 0.28; 0.9 | 0.6 | 0.27; 1.3 |
| 65 and + years | 0.12 | 0.09; 0.16 | 2.42 | 1.89; 3.10 | 1.06 | 0.76; 1.48 | 1.0 | 0.68; 1.4 | 0.5 | 0.34; 0.8 | 0.8 | 0.47; 1.2 | 0.9 | 0.51; 1.7 |
| Likelihood Ratio | test | p | test | p | test | p | test | p | test | p | test | p | test | p |
| | 717.44 | <0.001 | 158.37 | <0.001 | 47.84 | <0.001 | 12.8 | 0.1 | 81.3 | <0.001 | 94.0 | <0.001 | 80.0 | <0.001 |

| Age and sex | CORONAVIRUS OC43 | | CORONAVIRUS 229E | | PARAINFLUENZA 4 | | CORONAVIRUS NL63 | | PARAINFLUENZA1 | | CORONAVIRUS HKU1 | | PARAINFLUENZA 2 | |
|---|---|---|---|---|---|---|---|---|---|---|---|---|---|---|
| | OR | 95%CI | OR | 95%CI | OR | 95%CI | OR | 95%CI | OR | 95%CI | OR | 95%CI | OR | 95%CI |
| Man/Women | 0.97 | 0.71; 1.33 | 1.01 | 0.67; 1.44 | 1.01 | 0.7; 1.48 | 0.9 | 0.61; 1.3 | 1.9 | 1.07; 3.3 | 1.4 | 0.78; 2.4 | 1.8 | 0.82; 3.9 |
| Less than 1 year | 1.00 | –; – | 1.00 | –; – | 1.00 | –; – | 1.0 | –; – | 1.0 | –; – | 1.0 | –; – | 1.0 | –; – |
| 1–4 years | 1.18 | 0.71; 1.98 | 1.66 | 0.52; 5.23 | 1.33 | 0.71; 2.49 | 2.2 | 0.88; 5.5 | 5.1 | 1.51; 17.0 | 10.8 | 1.4; 82 | 0.6 | 0.2; 1.7 |
| 5–9 years | 0.51 | 0.19; 1.37 | 4.29 | 1.25; 14.76 | 1.46 | 0.64; 3.38 | 2.8 | 0.94; 8.5 | 1.7 | 0.28; 10.2 | 5.0 | 0.4; 55 | 1.1 | 0.28; 4.2 |
| 10–14 years | 1.13 | 0.38; 3.34 | 10.20 | 2.84; 36.60 | 2.69 | 1.03; 7.06 | 1.1 | 0.13; 9.1 | 4.6 | 0.77; 27.9 | – | –; – | 1.0 | 0.12; 8.0 |
| 15–24 years | 0.86 | 0.29; 2.52 | 11.70 | 3.57; 38.60 | 2.39 | 0.95; 5.96 | 7.7 | 2.68; 21.9 | 5.7 | 1.14; 28.6 | 21.6 | 2.4; 195 | 2.4 | 0.61; 9.5 |
| 25–44 years | 3.12 | 1.85; 5.24 | 12.60 | 4.44; 36.10 | 1.79 | 0.88; 3.64 | 8.8 | 3.66; 21.3 | 4.2 | 1.07; 16.2 | 27.2 | 3.6; 207 | 1.0 | 0.29; 3.5 |
| 45–64 years | 1.48 | 0.78; 2.82 | 16.50 | 5.77; 46.90 | 2.45 | 1.22; 4.92 | 6.1 | 2.4; 15.5 | 4.4 | 1.09; 17.6 | 19.1 | 2.4; 151 | 0.6 | 0.13; 3.0 |
| 65 and + years | 2.20 | 1.25; 3.87 | 12.50 | 4.34; 35.80 | 1.66 | 0.79; 3.47 | 5.1 | 1.99; 13.0 | 5.6 | 1.52; 20.9 | 7.3 | 0.8; 65 | 0.5 | 0.11; 2.5 |
| Likelihood Ratio | test | p | test | p | test | p | test | p | test | p | test | p | test | p |
| | 37.9 | <0.001 | 103.0 | <0.001 | 10.1 | 0.3 | 54.1 | <0.001 | 18.1 | <0.001 | 34.0 | <0.001 | 6.6 | 0.6 |

**Table 7. Risk of death associated with the presence of a respiratory virus.**

| VIRUS | RR | 95%CI |
|---|---|---|
| Respiratory syncytial virus | 0.53 | 0.32;0.86 |
| Rhinovirus | 0.9 | 0.58;1.38 |
| Metapneumovirus | 0.78 | 0.41;1.50 |
| Parainfluenza virus 3 | 0.54 | 0.23;1.22 |
| Bocavirus | 0.93 | 0.45;1.91 |
| Adenovirus | 0.95 | 0.46;1.94 |
| Enterovirus | _ | _;_ |
| Coronavirus OC43 | 1.1 | 0.41;2.97 |
| Coronavirus 229E | 3.1 | 1.54;6.24 |
| Parainfluenza virus 4 | 4.01 | 2.14;7.53 |
| Coronavirus NL63 | 0.82 | 0.20;3.29 |
| Parainfluenza virus 1 | 3.49 | 1.33;9.16 |
| Coronavirus HKU1 | 1.77 | 0.45;7.02 |
| Parainfluenza virus 2 | 3.32 | 0.86;12.80 |
| More than one virus | 0.47 | 0.24;0.94 |

2. The occurrence of outbreaks of adenovirus and syncytial virus starting in September 2022, which coincided with the reports from the United States, with an increase since August in the positive results and hospitalizations due to respiratory syncytial virus [19].

3. Surveillance bias for adenovirus infection in patients with acute hepatitis of unknown etiology [20].

Regarding sex, more samples were from males, mainly from children under 5 years of age. This finding has epidemiological and clinical importance since it has been documented that children have a greater risk of complications, such as bronchiolitis, increased hospitalization, elevated C-reactive protein levels and thrombocytosis, when they are diagnosed with respiratory syncytial virus infection [21]. Furthermore, bronchiolitis due to rhinovirus has been documented, and it was recently reported that rhinoviruses A and C were differentially associated with the identification of pathogenic bacteria in infants and school-age children, as well as with changes in the microbiota pattern in the presence of this virus. Patients with Rhinovirus C infection had a greater likelihood of exhibiting a *Moraxella*-dominated microbiota profile, while those with Rhinovirus A infection had a greater likelihood of exhibiting a *Haemophilus*-dominated profile [22].

Adenovirus coinfections, including those caused by other types of viruses, especially in children, have already been described, and no greater risk has been found [23–25]. In the present study, 16.9% of the overall samples were positive for 2 or more viruses, with a predominance in children under 5 years of age (25.0%). However, a study prior to the COVID-19 pandemic in the pediatric population in China revealed that this phenomenon occurred in 25% of samples, mainly between June and September [26].

With the development of technology and the affordability of sample processing for the identification of various viruses, the frequency of results positive for 2 or more simultaneous viruses is expected to increase.

The presence of simultaneous viruses did not represent an increased risk of death; conversely, it was observed to be a protective factor. In fact, there is controversy about the clinical severity of coinfections [27, 28]. In a study prior to the COVID-19 pandemic, clinical data,

**Table 8. Multivariate logistic regression analysis by age, sex and species of virus associated with death.**

| AGE AND SEX | RESPIRATORY SYNCYTIAL VIRUS* | | RHINOVIRUS* | | METAPNEUMOVIRUS* | | PARAINFLUENZA 3* | | BOCAVIRUS* | | ADENOVIRUS* | | ENTEROVIRUS* | |
|---|---|---|---|---|---|---|---|---|---|---|---|---|---|---|
| | OR | 95%CI | OR | 95%CI | OR | 95%CI | OR | 95%CI | OR | 95%CI | OR | 95%CI | OR | 95%CI |
| RSV (Yes/No)* | 1.0 | 0.6; 1.7 | 0.7 | 0.4; 1.1 | 0.9 | 0.4; 1.7 | 0.5 | 0.2; 1.2 | 1.3 | 0.6; 2.8 | 1.4 | 0.7; 3.0 | – | –; – |
| Man/Women | 1.4 | 0.91; 2.0 | 1.4 | 0.9; 2.0 | 1.4 | 0.9; 2.0 | 1.3 | 0.9; 2.0 | 1.4 | 0.9; 2.0 | 1.4 | 0.9; 2.0 | – | –; – |
| Less than 1 year | 1.0 | –; – | 1.0 | –; – | 1.0 | –; – | 1.0 | –; – | 1.0 | –; – | 1.0 | –; – | – | –; – |
| 1–4 years | 0.9 | 0.31; 2.4 | 0.9 | 0.3; 2.4 | 0.9 | 0.3; 2.3 | 0.9 | 0.3; 2.3 | 0.9 | 0.3; 2.3 | 0.9 | 0.3; 2.3 | – | –; – |
| 5–9 years | 1.1 | 0.27; 4.2 | 1.1 | 0.3; 4.4 | 1.1 | 0.3; 4.2 | 1.0 | 0.3; 4.1 | 1.1 | 0.3; 4.2 | 1.1 | 0.3; 4.1 | – | –; – |
| 10–14 years | 0.9 | 0.12; 8.0 | 1.0 | 0.1; 8.5 | 0.9 | 0.1; 7.8 | 1.0 | 0.1; 8.0 | 0.9 | 0.2; 8.0 | 1.0 | 0.1; 7.9 | – | –; – |
| 15–24 years | 3.1 | 0.87; 10.9 | 3.3 | 0.9; 11.6 | 3.1 | 0.9; 10.6 | 3.0 | 0.9; 10.3 | 3.1 | 0.9; 10.7 | 3.1 | 0.9; 10.7 | – | –; – |
| 25–44 years | 5.7 | 2.32; 13.8 | 6.0 | 2.5; 14.2 | 5.6 | 2.4; 13.3 | 5.6 | 2.4; 13.3 | 5.8 | 2.4; 13.6 | 5.7 | 2.4; 13.3 | – | –; – |
| 45–64 years | 8.1 | 3.39; 19.7 | 8.6 | 3.7; 20.1 | 8.1 | 3.5; 18.9 | 8.2 | 3.5; 19.1 | 8.2 | 3.5; 19.2 | 8.2 | 3.5; 19.2 | – | –; – |
| 65 and + years | 7.6 | 3.19; 18.2 | 8.0 | 3.5; 18.5 | 7.6 | 3.3; 17.5 | 7.6 | 3.3; 17.5 | 7.7 | 3.3; 17.7 | 7.6 | 3.3; 17.6 | – | –; – |
| Likelihood Ratio | test | $p$ | test | $p$ | test | $p$ | test | $p$ | test | $p$ | test | $p$ | test | $p$ |
| | 83.7 | <0.001 | 85.9 | <0.001 | 83.8 | <0.001 | 86.9 | <0.001 | 84.3 | <0.001 | 84.5 | <0.001 | – | – |

| AGE AND SEX | CORONAVIRUS OC43* | | CORONAVIRUS 229E* | | PARAINFLUENZA 4* | | CORONAVIRUS NL63* | | PARAINFLUENZA A1* | | CORONAVIRUS HKU1* | | PARAINFLUENZA 2* | |
|---|---|---|---|---|---|---|---|---|---|---|---|---|---|---|
| | OR | 95%CI | OR | 95%CI | OR | 95%CI | OR | 95%CI | OR | 95%CI | OR | 95%CI | OR | 95%CI |
| RSV (Yes/No)* | 0.8 | 0.3; 2.3 | 1.8 | 0.9; 3.9 | 4.1 | 2.0; 8.2 | 0.5 | 0.1; 2.2 | 3.3 | 1.1; 9.5 | 1.3 | 0.3; 5.4 | 3.9 | 0.9; 17.5 |
| Man/Women | 1.4 | 0.9; 2.0 | 1.4 | 0.9; 2.0 | 1.4 | 0.9; 2.0 | 1.4 | 0.9; 2.0 | 1.3 | 0.9; 2.0 | 1.4 | 0.9; 2.0 | 1.4 | 0.9; 2.0 |
| Less than 1 year | 1.0 | –; – | 1.0 | –; – | 1.0 | –; – | 1.0 | –; – | 1.0 | –; – | 1.0 | –; – | 1.0 | –; – |
| 1–4 years | 0.9 | 0.3; 2.3 | 0.9 | 0.3; 2.3 | 0.9 | 0.3; 2.3 | 0.9 | 0.3; 2.3 | 0.8 | 0.3; 2.3 | 0.9 | 0.3; 2.3 | 0.9 | 0.3; 2.3 |
| 5–9 years | 1.1 | 0.3; 4.1 | 1.1 | 0.3; 4.1 | 1.0 | 0.3; 4.1 | 1.1 | 0.3; 4.2 | 1.1 | 0.3; 4.1 | 1.1 | 0.3; 4.1 | 1.1 | 0.3; 4.1 |
| 10–14 years | 0.9 | 0.1; 7.8 | 0.9 | 0.1; 7.6 | 0.9 | 0.1; 7.4 | 1.0 | 0.1; 7.8 | 0.9 | 0.1; 7.6 | 1.0 | 0.1; 7.8 | 1.0 | 0.1; 7.8 |
| 15–24 years | 3.1 | 0.9; 10.6 | 3.0 | 0.9; 10.3 | 2.9 | 0.8; 10.2 | 3.1 | 0.9; 10.8 | 3.0 | 0.8; 10.3 | 3.1 | 0.9; 10.6 | 3.0 | 0.9; 10.4 |
| 25–44 years | 5.7 | 2.4; 13.4 | 5.4 | 2.3; 12.8 | 5.5 | 2.3; 12.9 | 5.8 | 2.4; 13.6 | 5.5 | 2.3; 13.0 | 5.6 | 2.4; 13.2 | 5.6 | 2.4; 13.3 |
| 45–64 years | 8.2 | 3.5; 19.0 | 7.7 | 3.3; 18.1 | 7.7 | 3.3; 18.1 | 8.3 | 3.6; 19.3 | 8.0 | 3.4; 18.5 | 8.1 | 3.5; 18.8 | 8.2 | 3.5; 19.1 |
| 65 and + years | 7.6 | 3.3; 17.6 | 7.3 | 3.2; 16.9 | 7.4 | 3.2; 17.2 | 7.7 | 3.3; 17.7 | 7.4 | 3.2; 17.0 | 7.6 | 3.3; 17.5 | 7.7 | 3.3; 17.7 |
| Likelihood Ratio | test | $p$ | test | $p$ | test | $p$ | test | $p$ | test | $p$ | test | $p$ | test | $p$ |
| | 83.9 | <0.001 | 103.0 | <0.001 | 95.3 | <0.001 | 84.7 | <0.001 | 87.3 | <0.001 | 83.8 | <0.001 | 86.0 | <0.001 |

laboratory findings, and type of care were analyzed among patients with coinfections, and no differences in severity were detected [26].

In contrast, during the study period, parainfluenza virus types 1, 2 and 4 showed the highest lethality in adults and older adults (average age, 44.6 years; 27% were over 65 years of age), it is generally accepted that parainfluenza virus 3 as well as respiratory syncytial virus infections are common in pediatric population, whereas parainfluenza virus 1 and 2 infections tend to be commoner in older persons; [29] even parainfluenza virus infection is frequently implicated in acute exacerbations of chronic obstructive pulmonary diseases [30] a situation that could be considered in the present study given that 27% of these were in older adults, with the limitation that the history of comorbidities is ignored [30]. Additionally, a meta-analysis carried out in 2018 on acute lower respiratory infection (ALRI) in children under 5 years of age highlighted that there is a global burden of parainfluenza attributable to approximately 13% of cases of ALRI infections, 4–14% of hospital admissions and 4% of child fatalities [31].

When ORs were calculated to determine the associations between age and the presence of different viral agents, it was found that in the of adenovirus infection was lower for older patients. A possible explanation is that most of the samples were taken between 2021 and 2022, while vaccination against COVID-19, which includes a vector virus, began in Mexico in December 2020.

Most of the viruses identified, such as respiratory syncytial virus, were observed in the pediatric population [32] and did not represent a risk of lethality, except for parainfluenza types 4 and 1. This difference remained when the models were adjusted, but an increased risk of death was also observed for older patients, and there was no access to other variables that were related to this outcome to establish any confounding factor. Previously, in a similar study with samples analyzed for respiratory viruses in Mexico, using RT–qPCR, the authors revealed that rhinovirus (33.0%) and human respiratory syncytial virus (30.8%) were the most common viruses among 14 noninfluenza respiratory viruses in 872 pharyngeal exudate samples from 2014 to 2015; [33] however, the sample size was smaller than that of our study. Similarly, another study carried out in Jalisco, Mexico, during the COVID-19 pandemic era (2021–2023) revealed that monoinfection represented the major cause of disease, principally SARS-CoV-2 and 7.9% for other viruses; of these, influenza virus was the most common (7.7%) [34]. This finding reinforces the importance of the study of other respiratory viruses and coinfections during in the pandemic stage. The rate of positivity for respiratory syncytial virus and rhinovirus was greater in 2020–2022 in most of the national territories.

Adenovirus has a wide range of clinical manifestations, not only at the respiratory level but also in other systems, due to the tropism that leads to complications or fatal outcomes [35].

This study had the following strengths: 1) the records used had laboratory-based results; 2) coinfections were tracked; 3) the IMSS provides services to more than half of Mexico's population, and the results obtained in this study may reflect the main respiratory viral types in this country; 4) the data collected came from patients treated in medical units at the 3 levels of care; 5) the number of records facilitated obtaining statistically significant results; and 6) most publications on respiratory viruses focus on the pediatric population, whereas this study covered all age groups.

This study also had several limitations. There was a differential bias for the collection of samples from children under 5 years of age, mainly due to active epidemiological surveillance in this population that attends daycare centers and a lack of vaccination against influenza. Although the risk of death increases with age, other potentially confounding variables are lacking (clinical manifestations, comorbidities). In addition, these samples were from a nonprobabilistic sampling of people negative for influenza or SARS-CoV-2, and due to the nature of the information system, clinical data other than the type of attention are not collected. Viral

identification is limited without considering the types or subtypes. Finally, in the statistical analysis, it was not possible to determine the risk of death based on time, since the sample records lacked the date of death and only included dates related to the sample, not to the patient's life events. The importance of the differential diagnosis of other respiratory viruses in samples with negative SARS-CoV-2 results becomes apparent when we observe the prevalence of the three main viruses identified in this study as well as their associations with severe cases and deaths.

In conclusion, respiratory syncytial virus and rhinovirus had the highest frequency and proportion of coinfections, while parainfluenza virus was associated with death. Therefore, laboratory-based epidemiological surveillance of respiratory viruses, including noninfluenza viruses and SARS-CoV-2, is essential for disease control and prevention.

## Supporting information

**S1 File. Sequence and working concentration of primers and probes.**
(DOCX)

## Acknowledgments

The authors thank all the participants in the viral respiratory disease epidemiological surveillance system, including treating physicians, epidemiologists, laboratory chemists, and nursing staff.

## Author Contributions

**Conceptualization:** Porfirio Felipe Hernández Bautista, David Alejandro Cabrera Gaytán.

**Data curation:** Julio Elias Alvarado Yaah, Bernardo Martínez Miguel.

**Formal analysis:** Porfirio Felipe Hernández Bautista, Alfonso Vallejos Parás, Julio Elias Alvarado Yaah.

**Investigation:** Porfirio Felipe Hernández Bautista, Alfonso Vallejos Parás, Alejandro Moctezuma Paz.

**Methodology:** Porfirio Felipe Hernández Bautista, Alejandro Moctezuma Paz.

**Project administration:** Clara Esperanza Santacruz Tinoco.

**Resources:** Clara Esperanza Santacruz Tinoco, Yu Mei Anguiano Hernández, Bernardo Martínez Miguel.

**Supervision:** Alejandro Moctezuma Paz, Yu Mei Anguiano Hernández, Nancy Sandoval Gutiérrez.

**Validation:** Alfonso Vallejos Parás, Lumumba Arriaga Nieto, Leticia Jaimes Betancourt.

**Visualization:** David Alejandro Cabrera Gaytán, Lumumba Arriaga Nieto.

**Writing – original draft:** David Alejandro Cabrera Gaytán, Alfonso Vallejos Parás, Lumumba Arriaga Nieto, Leticia Jaimes Betancourt.

**Writing – review & editing:** David Alejandro Cabrera Gaytán, Leticia Jaimes Betancourt.

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
