## [Decision Letter · Decision Letter 0]

13 Feb 2024

PONE-D-23-43288Respiratory viruses behavior: result of laboratory-based epidemiological surveillance.PLOS ONE

Dear Dr. Cabrera Gaytán,

Thank you for submitting your manuscript to PLOS ONE. After careful consideration, we feel that it has merit but does not fully meet PLOS ONE’s publication criteria as it currently stands. Therefore, we invite you to submit a revised version of the manuscript that addresses the points raised during the review process.

We look forward to receiving your revised manuscript.

Kind regards,

Yoon-Seok Chung

Academic Editor

PLOS ONE

Reviewers' comments:

Reviewer's Responses to Questions

**Comments to the Author**

1. Is the manuscript technically sound, and do the data support the conclusions?

Reviewer #1: Yes

Reviewer #2: Yes

2. Has the statistical analysis been performed appropriately and rigorously? 

Reviewer #1: Yes

Reviewer #2: I Don't Know

3. Have the authors made all data underlying the findings in their manuscript fully available?

Reviewer #1: Yes

Reviewer #2: Yes

4. Is the manuscript presented in an intelligible fashion and written in standard English?

Reviewer #1: Yes

Reviewer #2: Yes

5. Review Comments to the Author

Reviewer #1: In general, the authors described the epidemiological behavior of several common respiratory viruses other than influenza virus and SARS-CoV-2 in Mexico, according to multiplex PCR laboratory-based epidemiological surveillance.

These are my comments:

From the editorial point of view, the article flow and path was reasonable, however, I encountered several typo mistakes and grammar and writing issues throughout the text. Please double-check the whole text for revision.

- The title reflects the manuscript content.

- Abstract is adequate, describing the message of the experiments.

- Background: is adequately described.

The sentence: “Likewise, the adenovirus regains importance due to the mass vaccination with the biologicals that have been administered in Mexico to combat COVID-19, some of which contain adenovirus as a vector”. The authors mean that COVID-19 adenovirus vector-included vaccines caused to bring up this disease? Is it an exact issue or assumption? Please explain

- M&M: is reproducible.

Authors have not mentioned about the RNA extraction kit they used. If it covers all kind of RNA and DNA viruses? Also, any cDNA synthesis was performed before multiplex PCR?

- Results are described in proper way.

As there are age groups of infants and youngsters it is better to mention male and female instead of man/men and woman/women.

Figure 1 is too blurred and the range of colors is too close to distinct. Please change the color range.

Discussion: It is nicely organized and presented on the data justified, compared with other similar literatures.

In the part “The frequency by type of respiratory virus, other than influenza or SARS-CoV-2, was similar to the one presented in other parts of the world.[12,13,14,15,16]” …There are also some similar studies conducted in the middle east (Iran) that may add value to the context of the manuscript (PMID: 38093303, PMID: 36482963).

As I mentioned in introduction, I was concerned regarding adenovirus prevalence information. Authors should make a strong justification to relate the adenovirus prevalence to the COVID-19 adenovirus vector-included vaccines.

- References are up to date, just double-check for the consistency.

Overall comments:

The scientific content of the manuscript is really interesting and worth reporting for the health care system comparing the prevalence of several respiratory viruses all over the world after double-checking the whole text for the writing errors and applying corrections with critical consideration.

It has merit for publication after minor modification.

Reviewer #2: Summary of the manuscript:

This observational study aimed to characterize the epidemiological behavior of respiratory viruses from 2017 to 2023, with a focus on samples beyond SARS-CoV-2 due to increased interest during the COVID-19 pandemic. Using laboratory-based epidemiological surveillance, the study analyzed 4,632 samples through multiplex PCR. The most prevalent viruses were respiratory syncytial virus (26.3%) and rhinovirus (24.8%), primarily affecting children under 5 years. Parainfluenza virus 4 showed an age-adjusted odds ratio of 4.1 for death. In conclusion, respiratory syncytial virus and rhinovirus were common, while parainfluenza virus 4 was associated with fatalities.

Comments

1. There are already several reports as provided below to inform that such kind of work is already reported and that reduces the novelty of the current submission.

Reporting after Covid-19

Pedroza-Uribe IM, Vega Magaña N, Muñoz-Valle JF, Peña-Rodriguez M, Carranza-Aranda AS, Sánchez-Sánchez R, Venancio-Landeros AA, García-González OP, Zavala-Mejía JJ, Ramos-Solano M, Viera-Segura O and García-Chagollán M (2024) Beyond SARS-CoV-2: epidemiological surveillance of respiratory viruses in Jalisco, Mexico. Front. Public Health. 11:1292614. doi: 10.3389/fpubh.2023.1292614

Reporting before Covid-19

Fernandes-Matano L, Monroy-Muñoz IE, Angeles-Martínez J, Sarquiz-Martinez B, Palomec-Nava ID, Pardavé-Alejandre HD, Santos Coy-Arechavaleta A, Santacruz-Tinoco CE, González-Ibarra J, González-Bonilla CR, Muñoz-Medina JE. Prevalence of non-influenza respiratory viruses in acute respiratory infection cases in Mexico. PLoS One. 2017 May 3;12(5):e0176298. doi: 10.1371/journal.pone.0176298. PMID: 28467515; PMCID: PMC5415110.

2. Samples collected are not classified in terms of area and timelines. Demographic distribution of the samples used in the report may help further understanding.

3. As per the reported timelines of the survey, “Surveillance of respiratory viruses in Mexico has been carried out through the 2009-2020 Influenza Epidemiological Surveillance System and the Viral Respiratory Disease Epidemiological Surveillance System (ERV according to its initials in Spanish) (from 2020 to date) [10] as a non-probabilistic sampling of negative results for influenza or SARS-CoV-2.”. The reason for taking samples from 2017-2023 is not specified or justified.

4. Most of the samples are from Paediatric cohort and hence there are issues of variable susceptibility. The analysis may also be done in different groups to distinguish among the overall samples. Alternatively, only one cohort, i.e. paediatric may be considered for total duration from 2009 to till date in order to make the work and report a bit unique and a systematic one.

6. PLOS authors have the option to publish the peer review history of their article (what does this mean?). If published, this will include your full peer review and any attached files.

Reviewer #1: No

Reviewer #2: **Yes: **Dr Vijay Nema

---

## [Author Response · Author response to Decision Letter 0]

28 Feb 2024

Reviewer #1.

In general, the authors described the epidemiological behavior of several common respiratory viruses other than influenza virus and SARS-CoV-2 in Mexico, according to multiplex PCR laboratory-based epidemiological surveillance.

These are my comments:

From the editorial point of view, the article flow and path was reasonable, however, I encountered several typo mistakes and grammar and writing issues throughout the text. Please double-check the whole text for revision.

- The title reflects the manuscript content.

- Abstract is adequate, describing the message of the experiments.

- Background: is adequately described.

The sentence: “Likewise, the adenovirus regains importance due to the mass vaccination with the biologicals that have been administered in Mexico to combat COVID-19, some of which contain adenovirus as a vector”. The authors mean that COVID-19 adenovirus vector-included vaccines caused to bring up this disease? Is it an exact issue or assumption? Please explain.

A = This is an assumption: “Likewise, the adenovirus regains importance due to the mass vaccination with the biologicals that have been administered in Mexico to combat COVID-19, some of which contain adenovirus as a vector; that would probably confer protection for the adenovirus”. It was decided to delete the paragraph to avoid confusion.

- M&M: is reproducible.

Authors have not mentioned about the RNA extraction kit they used. If it covers all kind of RNA and DNA viruses? Also, any cDNA synthesis was performed before multiplex PCR?

A = This section was added.

Laboratory analysis of samples.

The SuperScript™ III Platinum® One-Step RT‒qPCR System Kit (Invitrogen, Carlsbad, California, USA. Catalog: 12574035) was used with a 7500 Fast Real-Time PCR System (Applied Biosystems®, Foster City, California, USA) to amplify viral genetic material. Primers and probes were used for each of the following viruses: metapneumovirus, respiratory syncytial virus, human parainfluenza viruses 1–4, betacoronavirus 1, human coronavirus (OC43, HKU1, 229E, NL63), rhinovirus, enterovirus, adenovirus and bocavirus. The human RNAse P (RP) gene was used as an internal control. The samples were evaluated in uniplex reactions with the following reaction mixture: 12.5 μL of 2x reaction mix, 0.5 μL of each primer and probe, 1 μL of enzyme, 5.5 μL of RNase-free water, and 5 μL of total nucleic acid. The following thermocycling conditions were used: one cycle of 45°C for 10 min and 95°C for 10 min and 45 cycles of 95°C for 15 sec and 55°C for 1 min. DNA or RNA lyophilizates (AmpliRun® Vircell; Granada–Spain) were used as the positive controls for all evaluated viruses. Samples that presented amplification for any of the viral markers plus the internal control were considered positive. Samples without amplification of the viral markers but with amplification of the internal control were considered negative.

- Results are described in proper way.

As there are age groups of infants and youngsters it is better to mention male and female instead of man/men and woman/women.

A = The changes were made.

Figure 1 is too blurred and the range of colors is too close to distinct. Please change the color range.

A = The changes were made.

Discussion: It is nicely organized and presented on the data justified, compared with other similar literatures.

In the part “The frequency by type of respiratory virus, other than influenza or SARS-CoV-2, was similar to the one presented in other parts of the world.[12,13,14,15,16]” …There are also some similar studies conducted in the middle east (Iran) that may add value to the context of the manuscript (PMID: 38093303, PMID: 36482963).

A = The changes were made. Suggested reference added.

The frequency by type of respiratory virus, other than influenza or SARS-CoV-2, was similar to the one presented in other parts of the world.[12-16] Yet, it contrasts with a study carried out in China during the COVID-19 pandemic, since between 2020 and 2021, the results for adenovirus decreased considerably, [17] also in Iran adenovirus had the lowest prevalence (0.1%) whereas in our 2020 study compared to the previous year an increase was witnessed, which was stressed in 2022.

As I mentioned in introduction, I was concerned regarding adenovirus prevalence information. Authors should make a strong justification to relate the adenovirus prevalence to the COVID-19 adenovirus vector-included vaccines.

A = It was decided to delete the paragraph to avoid confusion. 

- References are up to date, just double-check for the consistency.

A = The changes were made.

Overall comments:

The scientific content of the manuscript is really interesting and worth reporting for the health care system comparing the prevalence of several respiratory viruses all over the world after double-checking the whole text for the writing errors and applying corrections with critical consideration.

It has merit for publication after minor modification.

Reviewer #2. 

Summary of the manuscript:

This observational study aimed to characterize the epidemiological behavior of respiratory viruses from 2017 to 2023, with a focus on samples beyond SARS-CoV-2 due to increased interest during the COVID-19 pandemic. Using laboratory-based epidemiological surveillance, the study analyzed 4,632 samples through multiplex PCR. The most prevalent viruses were respiratory syncytial virus (26.3%) and rhinovirus (24.8%), primarily affecting children under 5 years. Parainfluenza virus 4 showed an age-adjusted odds ratio of 4.1 for death. In conclusion, respiratory syncytial virus and rhinovirus were common, while parainfluenza virus 4 was associated with fatalities.

Comments

1. There are already several reports as provided below to inform that such kind of work is already reported and that reduces the novelty of the current submission.

Reporting after Covid-19

Pedroza-Uribe IM, Vega Magaña N, Muñoz-Valle JF, Peña-Rodriguez M, Carranza-Aranda AS, Sánchez-Sánchez R, Venancio-Landeros AA, García-González OP, Zavala-Mejía JJ, Ramos-Solano M, Viera-Segura O and García-Chagollán M (2024) Beyond SARS-CoV-2: epidemiological surveillance of respiratory viruses in Jalisco, Mexico. Front. Public Health. 11:1292614. doi: 10.3389/fpubh.2023.1292614

Reporting before Covid-19

Fernandes-Matano L, Monroy-Muñoz IE, Angeles-Martínez J, Sarquiz-Martinez B, Palomec-Nava ID, Pardavé-Alejandre HD, Santos Coy-Arechavaleta A, Santacruz-Tinoco CE, González-Ibarra J, González-Bonilla CR, Muñoz-Medina JE. Prevalence of non-influenza respiratory viruses in acute respiratory infection cases in Mexico. PLoS One. 2017 May 3;12(5):e0176298. doi: 10.1371/journal.pone.0176298. PMID: 28467515; PMCID: PMC5415110.

A = The changes were made. Suggested reference added.

2. Samples collected are not classified in terms of area and timelines. Demographic distribution of the samples used in the report may help further understanding.

A = Data is added in the study to complement this aspect. We added Figure 2.

3. As per the reported timelines of the survey, “Surveillance of respiratory viruses in Mexico has been carried out through the 2009-2020 Influenza Epidemiological Surveillance System and the Viral Respiratory Disease Epidemiological Surveillance System (ERV according to its initials in Spanish) (from 2020 to date) [10] as a non-probabilistic sampling of negative results for influenza or SARS-CoV-2.”. The reason for taking samples from 2017-2023 is not specified or justified.

A = Added text adding explanation.

Surveillance of respiratory viruses in Mexico has been carried out through the 2009-2020 Influenza Epidemiological Surveillance System[9] and the Viral Respiratory Disease Epidemiological Surveillance System (ERV according to its initials in Spanish) (from 2020 to date) [10] as a non-probabilistic sampling of negative results for influenza or SARS-CoV-2; in both periods with different operational case definitions, but in both in a syndromatic manner to identify mild and severe cases of acute respiratory disease. In this sense, samples were analyzed from years prior to the COVID-19 pandemic, as well as during it.

4. Most of the samples are from Paediatric cohort and hence there are issues of variable susceptibility. The analysis may also be done in different groups to distinguish among the overall samples. Alternatively, only one cohort, i.e. paediatric may be considered for total duration from 2009 to till date in order to make the work and report a bit unique and a systematic one.

A = Data is available from 2017. Likewise, the protocol submitted for evaluation by the Committee was approved for the period from 2017 to 2023.

---

## [Decision Letter · Decision Letter 1]

24 Apr 2024

PONE-D-23-43288R1Respiratory virus behavior: results of laboratory-based epidemiological surveillance.PLOS ONE

Dear Dr. Cabrera Gaytán,

Thank you for submitting your manuscript to PLOS ONE. After careful consideration, we feel that it has merit but does not fully meet PLOS ONE’s publication criteria as it currently stands. Therefore, we invite you to submit a revised version of the manuscript that addresses the points raised during the review process.

We look forward to receiving your revised manuscript.

Kind regards,

Yoon-Seok Chung

Academic Editor

PLOS ONE

Journal Requirements:

Reviewers' comments:

Reviewer's Responses to Questions

**Comments to the Author**

1. If the authors have adequately addressed your comments raised in a previous round of review and you feel that this manuscript is now acceptable for publication, you may indicate that here to bypass the “Comments to the Author” section, enter your conflict of interest statement in the “Confidential to Editor” section, and submit your "Accept" recommendation.

Reviewer #1: All comments have been addressed

Reviewer #3: (No Response)

2. Is the manuscript technically sound, and do the data support the conclusions?

Reviewer #1: Yes

Reviewer #3: Yes

3. Has the statistical analysis been performed appropriately and rigorously? 

Reviewer #1: I Don't Know

Reviewer #3: Yes

4. Have the authors made all data underlying the findings in their manuscript fully available?

Reviewer #1: No

Reviewer #3: Yes

5. Is the manuscript presented in an intelligible fashion and written in standard English?

Reviewer #1: Yes

Reviewer #3: Yes

6. Review Comments to the Author

**Reviewer #1:** The authors have addressed most of the comments.

There are just 3 things to address before acceptance:

1- Figure 1 is still blurring and I think if the authors cannot correct it, Journal editorial office may help to make it sharp.

2- Also I noticed some changes in the height of bars in new Figure 1 compared to old one. If the authors have made any changes in sample number and statistical analysis it is substantial to be mentioned to Editor.

3- In the new paragraph under “Laboratory analysis of samples” title; please add information about the RNA extraction kit used. If it covers all kind of RNA and DNA viruses?

**Reviewer #3**: This study analyzed laboratory-based respiratory virus surveillance data from 2017 to 2023 in Mexico, before and during the COVID-19 pandemic. Similar studies on respiratory virus infections have been published nationally and regionally.

However, this study has strengths in analyzing laboratory test results from patients of all ages in Mexico, excluding influenza and COVID-19 infections, to analyze the national respiratory virus epidemic pattern over time.

In laboratory-based epidemiological surveillance, authors can discuss hypotheses and inferences about differences in characteristics by age and non-clinical type of respiratory virus, and the possibility of co-infection, or suggest the possibility of follow-up research.

My opinions are as follows:

- indicate whether surveillance data from IESS (Influenza Epidemiological Surveillance System) and ERV (Respiratory Virus Epidemiology) are methodologically the same, and whether there are differences in period and surveillance system.

- The increase in ADV for children under 5 years of age presented in lines 189-190 is not clearly visible due to resolution issues in Figure 1.

- Since there was no epidemic of pediatric hepatitis caused by ADV, the reason for analyzing ADV and co-infections in Table 3 is unclear.

- The reason for analyzing enterovirus together with respiratory viruses is not specified.

- Explanation of the respiratory disease diagnosis algorithm in the Results section and clarification of the 1.6% of other respiratory viruses mentioned in Table 3 are needed.

- In the time series representation in Figure 1, adding years along with the months as separate lines on the x-axis from 2017 to January 2023 improves data interpretation.

- Indicating in the table captions the statistical methods used in Tables 6, 7, 8, and 9 will improve interpretation of the data. These are logistic regression analysis, multivariate logistic regression analysis, and multivariate analysis that fit the table.

- Reference 29 is about human coronaviruses, but the comparison with parainfluenza viruses and its relevance to the literature on parainfluenza virus mortality needs to be clarified a little more.

Minor terminology needs correction.

- Combine the two periods at the end of line 92 into one.

- Line 177: Ensure consistency of decimal places by maintaining one decimal place.

- Line 217: In the Results section, it is recommended to use "95%IC" instead of "95%CI" in Table 8 to be consistent with Tables 6, 7, and 9.

- Curiosity: I'm curious what the "a" in "1a4year" in all tables represents.

- Lines 167, 174, 175, 176: For consistency, use commas for numbers over 1000.

- Including line 173, it appears that the "O" for the OC43 subtype appears as the number 0 throughout the journal.

Confirm with minor review

- Clarification is needed regarding the mention of betacoronavirus 1, which is not the focus of this study, in line 132.

- Line 182: Check the number of patients for “86”.

- Consider expressing the unit as ‘fatality rate (%)’ in Table 5.

- Check once more the calculated values of 34.5% for the entire sample in line 269 and 18.8-36.2% in line 273.

7. PLOS authors have the option to publish the peer review history of their article (what does this mean?). If published, this will include your full peer review and any attached files.

Reviewer #1: No

Reviewer #3: No

---

## [Author Response · Author response to Decision Letter 1]

30 May 2024

Reviewer #1:

The authors have addressed most of the comments. There are just 3 things to address before acceptance:

1- Figure 1 is still blurring and I think if the authors cannot correct it, Journal editorial office may help to make it sharp.

A = The change request was made.

2- Also I noticed some changes in the height of bars in new Figure 1 compared to old one. If the authors have made any changes in sample number and statistical analysis it is substantial to be mentioned to Editor.

A = The observation is correct and pertinent. We realized that there was an error when selecting the virus data over time; that is why they were modified. We apologize for this.

3- In the new paragraph under “Laboratory analysis of samples” title; please add information about the RNA extraction kit used. If it covers all kind of RNA and DNA viruses?

A = We added lines and “S1. Sequence and working concentration of primers and probes”.

Reviewer #3:

This study analyzed laboratory-based respiratory virus surveillance data from 2017 to 2023 in Mexico, before and during the COVID-19 pandemic. Similar studies on respiratory virus infections have been published nationally and regionally.

However, this study has strengths in analyzing laboratory test results from patients of all ages in Mexico, excluding influenza and COVID-19 infections, to analyze the national respiratory virus epidemic pattern over time.

In laboratory-based epidemiological surveillance, authors can discuss hypotheses and inferences about differences in characteristics by age and non-clinical type of respiratory virus, and the possibility of co-infection, or suggest the possibility of follow-up research.

My opinions are as follows:

- Indicate whether surveillance data from IESS (Influenza Epidemiological Surveillance System) and ERV (Respiratory Virus Epidemiology) are methodologically the same, and whether there are differences in period and surveillance system.

A = Both are the same, in 2020 the name was changed and new COVID-19 variables were included (see line 88-89).

- The increase in ADV for children under 5 years of age presented in lines 189-190 is not clearly visible due to resolution issues in Figure 1.

A = The observation is correct; the statement was removed.

- Since there was no epidemic of pediatric hepatitis caused by ADV, the reason for analyzing ADV and co-infections in Table 3 is unclear.

A = We eliminated Table 3.

- The reason for analyzing enterovirus together with respiratory viruses is not specified.

A = Enterovirus is included in the RT-qPCR (respiratory virus) diagnostic panel (FilmArray™).

- Explanation of the respiratory disease diagnosis algorithm in the Results section and clarification of the 1.6% of other respiratory viruses mentioned in Table 3 are needed.

A = An explanation was included in the paragraph. The value provided is not linked to table 3. Also we eliminated that table.

- In the time series representation in Figure 1, adding years along with the months as separate lines on the x-axis from 2017 to January 2023 improves data interpretation.

A = The change request was made.

- Indicating in the table captions the statistical methods used in Tables 6, 7, 8, and 9 will improve interpretation of the data. These are logistic regression analysis, multivariate logistic regression analysis, and multivariate analysis that fit the table.

A = The request to add the type of analysis was included.

- Reference 29 is about human coronaviruses, but the comparison with parainfluenza viruses and its relevance to the literature on parainfluenza virus mortality needs to be clarified a little more.

A = The lines 285-294 were modified and the reference.

Minor terminology needs correction.

- Combine the two periods at the end of line 92 into one.

A = The change request was made.

- Line 177: Ensure consistency of decimal places by maintaining one decimal place.

A = The change request was made.

- Line 217: In the Results section, it is recommended to use "95%IC" instead of "95%CI" in Table 8 to be consistent with Tables 6, 7, and 9.

A = The change request was made.

- Curiosity: I'm curious what the "a" in "1a4year" in all tables represents.

A = An apology for the carelessness in the translation. The change request was made.

- Lines 167, 174, 175, 176: For consistency, use commas for numbers over 1000.

A = The change request was made.

- Including line 173, it appears that the "O" for the OC43 subtype appears as the number 0 throughout the journal.

A = The change request was made in the manuscript and tables.

- Clarification is needed regarding the mention of betacoronavirus 1, which is not the focus of this study, in line 132.

A = The change request was made.

- Line 182: Check the number of patients for “86”.

A = The change request was made.

- Consider expressing the unit as ‘fatality rate (%)’ in Table 5.

A = The change request was made. In the same way, it is homologated to a decimal.

- Check once more the calculated values of 34.5% for the entire sample in line 269 and 18.8-36.2% in line 273.

A = The lines were modified.

---

## [Decision Letter · Decision Letter 2]

4 Jul 2024

Respiratory virus behavior: results of laboratory-based epidemiological surveillance.

PONE-D-23-43288R2

Dear Dr. Cabrera Gaytán,

We’re pleased to inform you that your manuscript has been judged scientifically suitable for publication and will be formally accepted for publication once it meets all outstanding technical requirements.

Kind regards,

Yoon-Seok Chung

Academic Editor

PLOS ONE

Reviewers' comments:

Reviewer's Responses to Questions

**Comments to the Author**

1. If the authors have adequately addressed your comments raised in a previous round of review and you feel that this manuscript is now acceptable for publication, you may indicate that here to bypass the “Comments to the Author” section, enter your conflict of interest statement in the “Confidential to Editor” section, and submit your "Accept" recommendation.

Reviewer #1: (No Response)

Reviewer #3: All comments have been addressed

2. Is the manuscript technically sound, and do the data support the conclusions?

Reviewer #1: Yes

Reviewer #3: Yes

3. Has the statistical analysis been performed appropriately and rigorously? 

Reviewer #1: I Don't Know

Reviewer #3: Yes

4. Have the authors made all data underlying the findings in their manuscript fully available?

Reviewer #1: Yes

Reviewer #3: Yes

5. Is the manuscript presented in an intelligible fashion and written in standard English?

Reviewer #1: Yes

Reviewer #3: Yes

6. Review Comments to the Author

Reviewer #1: The authors have addressed all the comments except for 1 item.

In the paragraph “Laboratory analysis of samples” and in the supporting file S1, despite more information on PCR, there is still lack of information about the nucleic acid extraction kit used. If it covers all kind of RNA and DNA viruses? It is important to mention this

The manuscript has merit for publication after minor modification.

Reviewer #3: (No Response)

7. PLOS authors have the option to publish the peer review history of their article (what does this mean?). If published, this will include your full peer review and any attached files.

Reviewer #1: No

Reviewer #3: No

---

## [Editor Report · Acceptance letter]

7 Aug 2024

PONE-D-23-43288R2 

PLOS ONE

Dear Dr. Cabrera Gaytán, 

I'm pleased to inform you that your manuscript has been deemed suitable for publication in PLOS ONE. Congratulations! Your manuscript is now being handed over to our production team.

Kind regards, 

on behalf of

Dr. Yoon-Seok Chung 

Academic Editor

PLOS ONE